# Substitution of H Atoms in Unsaturated (Vinyl-Type) Carbocations by Cl or O Atoms

**DOI:** 10.3390/ijms241310734

**Published:** 2023-06-27

**Authors:** Evgenii S. Stoyanov, Irina Yu. Bagryanskaya, Irina V. Stoyanova

**Affiliations:** N.N. Vorozhtsov Institute of Organic Chemistry, Siberian Branch of Russian Academy of Sciences, 630090 Novosibirsk, Russia; bagryan@nioch.nsc.ru (I.Y.B.); stoyanova@nioch.nsc.ru (I.V.S.)

**Keywords:** vinyl carbocations, carborane salts, IR spectroscopy

## Abstract

Introduction of Cl and O atoms into C_4_-vinyl carbocations was studied by X-ray diffraction analysis and IR spectroscopy. Chlorine atoms are weak electron acceptors in ordinary molecules but, within vinyl carbocations, manifest themselves as strong electron donors that accept a positive charge. The attachment of a Cl atom directly to a C=C bond leads to an increase in the e-density on it, exceeding that of the common double bond. The positive charge should be concentrated on the Cl atom, and weak δ^−^ may appear on the C=C bond. More distant attachment of the Cl atom, e.g., to a C atom adjacent to the C=C bond, has a weaker effect on it. If two Cl atoms are attached to the C_γ_ atom of the vinyl cation, as in Cl_2_C_γ_C_δ_HC_α_HCH_3_, then the cation switches to the allyl type with two practically equivalent and almost uncharged C_γ_C_δ_C_α_ bonds. When such a strong nucleophile as the O atom is introduced into the carbocation, a protonated ester molecule with a C–O(H^+^)–C group and a C=C bond forms. Nonetheless, in the future, there is still a possibility of obtaining carbocations with a non-protonated C–O–C group.

## 1. Introduction

Until recently, vinyl carbocations have been thought to be highly reactive and, therefore, difficult to investigate [1,2,3], although recent studies showed that their reactivity is exaggerated [4,5,6]. The first vinyl cations whose salts were isolated in pure form and analyzed by X-ray diffraction and NMR methods were stabilized by substituents such as methyl, cycloalkyl, phenyl, and R_3_Si groups [2,7,8,9]. Stabilization occurs due to a scattering of the charge of the cation over atoms of these substituents, which act as electron donors. It has turned out that the supply of electron density from substituents increases electron density on the C=C bond so much that it approaches triple-bond status [7]. Later, salts of unstabilized vinyl cations C_3_H_7_^+^ and C_4_H_9_^+^ were obtained [10,11,12]. In them, the positive charge is concentrated mainly on the C=C bond, thereby greatly reducing its CC stretch frequency to that corresponding to one-and-a-half-bond status. Interaction of cations with surrounding anions with the formation of contact ion pairs or the introduction of Cl substituents also contributes to better scattering of the positive charge and its decrease in C=C bonds. The use of carborane superacids has made it possible to obtain and study solid salts of unstabilized vinyl and acetylene carbocations, which are stable at room and elevated temperatures [10,11,12,13].

In the present work, two goals were set: (1) to establish how the nature of the C=C bond of a vinyl cation is affected by bonding of its various C atoms to one weak nucleophile, a Cl atom, or two Cl atoms, and (2) how the carbocation changes when a stronger nucleophile (an oxygen atom) is introduced. The salts of carbocations were studied by X-ray crystallography and IR spectroscopy. As a counterion, the undecachlorocarborane anion, CHB_11_Cl_11_^−^, was chosen (Appendix A) because its extreme stability and low basicity promote the formation of stable ionic salts with highly acidic cations [14]. In what follows, this anion is denoted as {Cl_11_^−^}.

## 2. Results

Crystals of a salt of the dichlorovinyl cation, C_4_H_5_Cl_2_^+^{Cl_11_^−^}, were obtained as follows: a few drops of a dichloromethane (DCM) solution of CH_3_C(CH_2_Cl)_3_ was introduced into a freshly prepared solution of acid H{Cl_11_} (3–5 mg) in DCM (~1 mL) in such an amount that the molar ratio of CH_3_C(CH_2_Cl)_3_ to H{Cl_11_} approached 1.0. The solution first turned cloudy; then, within 15 min, it darkened strongly and became transparent. That is, sequential reactions took place. A drop of the solution was evaporated on the surface of a crystal of the ATR accessory. The recorded ATR IR spectrum contained strong bands of C=C stretch vibrations at 1565 cm^−1^, which are characteristic of vinyl cat ions (Figure 1), and did not contain absorption bands of chloronium cation (CH_2_Cl)_2_Cl^+^, which forms upon dissolution of the acid H{Cl_11_} in DCM [15]. It was found that, when interacting with chloroalkanes, the chloronium cation, as its neutral analog CH_2_Cl{Cl_11_}, takes away a chlorine atom from them, turning into DCM [10,11]. Therefore, when mixing solutions at the first stage, the reaction of dechlorination of chloroalkane should occur, giving rise to chlorinated carbocation C_5_H_7_Cl_2_^+^ (Equation (1)), which is unstable and spontaneously decomposes to the vinyl carbocation (with the band of C=C stretch frequency at 1565 cm^−1^, Figure 1) with a release of HCl:CH_3_C(CH_2_Cl)_3_ + (CH_2_Cl)_2_Cl^+^ → {C_5_H_7_Cl_2_^+^} + 2 CH_2_Cl_2_.unstable(1)

Incubation the solution for 2 days led to the growth of crystals from it. Their IR spectrum differed from the spectrum of products arising in the initial solution (Figure 1) in that it lacked the bands of the double C=C^+^ bonds at ≥1490 cm^−1^ [12]. X-ray diffraction analysis of the crystals indicated that this is an ionic salt with discrete {Cl_11_^−^} anions and two types of crystallographically independent 1,1-dichlorobutylene cations that are very similar in geometry (Appendix A; the structure of cations with averaged CC distances is given in Figure 2). Their two Cl atoms and four C atoms were actually in the same plane with an average deviation from the least-squares plane of 0.029 Å for one cation and 0.075 Å for the other. Lengths of the two CC bonds (averaged 1.34 Å) were close to aromatic, and angles C1C2C3 and C2C3C4 were close to 120° (Figure 2). This means that the atoms C1, C2, and C3 had sp^2^ hybridization and belonged to groups CCl_2_, CH, and CH, respectively. It is expected that the positive charge is localized mainly on the atoms of the CCl_2_ group, which forms the C1=C2 double bond. Nevertheless, the electron density is evenly distributed over two equivalent bonds of the C1C2C3 group, thereby equalizing their multiplicity to one and a half.

The IR spectrum of the 1,1-dichlorobutylene cation showed two intense bands of CC stretch vibrations at 1470 and 1378 cm^−1^, which may belong to ν_as_ and ν_as_ frequencies of the 
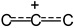
 group. These frequencies are higher than those of allyl cation 
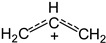
 (1303 and 1265 cm^−1^), indicating a decrease in the positive charge on this group [12], owing to transfer of the charge to Cl atoms. The spectrum also contained two bands of symmetric and asymmetric C–Cl stretch vibrations at 700 and 654 cm^−1^ in accordance with the presence of two C–Cl bonds at one carbon atom in the cation. 

Previously, we researched the interaction of vinyl carbocations, C_3_H_5_^+^ and C_4_H_7_^+^, with water molecules in solutions of their salts in C_6_HF_5_ [16]. One of the objectives of the current work is to check how the chlorination of vinyl carbocations affects their interaction with water molecules. For their preparation, cyclobutylchloronium (CH_2_CH_2_)_2_Cl^+^ and divinylchloronium (CH_2_=CH)_2_Cl^+^ salts were used. The salt of cyclobutylchloronium is poorly soluble in C_6_HF_5_ if it contains trace amounts of water (the solvent was pre-dried with molecular sieves). Keeping this saturated solution over hexane vapor led to the isolation of 2–3 crystals from it after a few days. The center of each crystal contained a disordered crystallite, around which a regular crystal grew. This finding indirectly indicates that the crystals are formed by one of the products of the reactions occurring in solution. A piece of its outer part that was broken off from the crystal showed a good diffraction pattern. X-ray analysis showed that the crystal consisted of discrete {Cl_11_^−^} anions and two crystallographically independent cations (Appendix A). The cations were protonated ether L_1_–O(H^+^)–L_2_, in which both L_1_ and L_2_ contained four carbon atoms each (Figure 3). Its chloropropane moiety L_1_ contained single C–C bonds and a terminal CH_2_Cl group. The second group, L_2_, was isobutylene with one short CC bond of 1.274 and 1.278 Å for two crystallographically independent cations, which corresponded to the C=C double bond. All its four C atoms and oxygen atom deviated from the least-squares plane by 0.011 and 0.010 Å for both cations with the C–C–O angle close to 120°, confirming that it is an isobutylene group: (CH_3_)_2_C=CH–. Hereafter, this cation is referred to as **I**.

Using divinylchloronium salt (CH_2_=CH)_2_Cl^+^{Cl_11_^−^} (preparation is given in [13]), another similar cation was obtained. A saturated solution of this salt in C_6_HF_5_ was kept over hexane vapor, and, after 2 weeks, small crystals appeared. The first selected crystal (0.1 mm) and the second smaller one (0.05 mm) yielded no diffraction pattern. A third very small crystal (0.005 × 0.01 × 0.30 mm) showed a diffraction pattern, and an X-ray pattern was obtained with long exposure. Such a small crystal size (less than 0.005 mm in one of the dimensions) led to the interpretation of the structure with a poor R_f_ factor (11.8%). Therefore, we do not discuss the geometric parameters of the cation. Nonetheless, the topology of the cation and its geometry were determined (Figure 4). It turned out to be protonated ether L′_1_–O(H^+^)–L_2_, with exactly the same isobutylene group L_2_ as in cation **I**. The L′_1_ group was chain butylene, which differed from the corresponding L_1_ group in **I** in that it contained two Cl atoms attached to atoms C7 and C8. All four atoms, together with the C6 atom, were in the same plane, with the mean deviation from the least-squares plane of 0.121 Å. This means that atoms C7 and C8 had sp^2^ hybridization and formed CCl and CHCl groups, respectively, with a C7=C8 double bond. The C5C6C7 angle at 108° was close to tetrahedral, i.e., the C6 atom formed a CH_2_ group. Thus, the scheme of the structure of this cation was established (Figure 4); in the text below, we designate it as cation **II**. 

## 3. Discussion

Let us examine how the attachment of a Cl atom to different carbon atoms of the butylene cation affects the nature of its CC bonds, by means of the known data on chlorobutylene carbocations [12]. In symmetrical vinyl cation *a* (Figure 1), which does not contain Cl atoms, the positive charge is located mainly on the C=C double bond. Therefore, its C=C stretching frequency (1490 cm^−1^) is much lower than that of the neutral 2-butene molecule (1645 cm^−1^). Replacing the H atom in the –CH=C^+^– group of this cation with a Cl atom (cation *b*) leads to such a substantial supply of electron density from the Cl atom to the C=C bond that the C=C stretch frequency increases by 220 cm^−1^ (up to 1710 cm^−1^) exceeding that of neutral 2-butene. Obviously, the Cl atom completely accepts the positive charge, extinguishing it on the C=C bond. If the Cl atom is attached to the C_α_ atom adjacent to the C=C bond (cation *c*), then its effect on the weakening of the charge on it is weaker (its effect on reducing its charge is weaker) (C=C stretch frequency increases by 190 cm^−1^). Attachment of the Cl atom to the C_γ_ atom of the terminal C=C bond (cation *d*) leads to a decrease in the multiplicity of the C_γ_C_β_ bond owing to the partial transfer of π-electron density to the C_β_C_α_ bond (according to NMR data). Lastly, the addition of the second Cl atom to the terminal C_γ_ atom with sp^2^ hybridization (cation **I** under study) leads to the alignment of multiplicities of C_γ_C_β_ and C_β_C_α_ bonds and the transfer of the positive charge to two Cl atoms. This event leads to an increase in the frequencies of C_γ_C_β_C_α_ stretch vibrations as compared to the unstabilized allyl cation (CH_2_CHCH_2_)^+^ (by 170 cm^−1^), bringing them closer to aromatic ones in neutral molecules.

Thus, in neutral CH_2_=CCl-R molecules, the chlorine atom has properties of an electron acceptor, slightly lowering the C=C stretch frequency by 20 cm^−1^ [17], whereas, in carbocations, it acts as an electron donor, extinguishing the positive charge on the C=C bond to a greater extent with decreasing distance to the double C=C bond. The chlorine atom also contributes to the delocalization of π-electron density to the neighboring CC bond.

We do not have the necessary data to discuss a possible mechanism of formation of C_4_-cation C_4_H_5_Cl_2_^+^ from C_5_-cation C_5_H_7_Cl_2_^+^ emerging at the first stage of the reaction, according to Equation (1). This reaction requires spontaneous cleavage of the C–C bond in the C_5_H_7_Cl_2_^+^ cation with the formation of the C_4_-cation and other unidentified products and, therefore, requires additional research. 

Cations **I** and **II** are products of the interaction of chlorinated vinyl carbocations with water molecules. In the case of non-chlorinated vinyl cations C_3_H_5_^+^ and C_4_H_7_^+^, their interaction with water molecules in solutions of their salts in C_6_HF_5_ proceeds in three stages as the water content increases (Figure 2) [16]: (i) adduct **1** arises; (ii) an alcohol molecule is formed from it, which attaches to the vinyl cation giving rise to adduct **2**; (iii) next, it transforms into proton disolvate **3**. According to quantum chemical calculations, adduct **2** is unstable and is likely to transform into protonated ether cation **4**. Nevertheless, in solutions in C_6_HF_5_, it is cation **3** that is the final product.

On the other hand, if cyclobutylchloronium or divinylchloronium salts are dissolved in C_6_HF_5_, then the chlorinated vinyl cations arising from them—by subsequently interacting with H_2_O—form precisely the protonated esters **I** and **II**, respectively. The formation of cation **I** from cyclobutylchloronium occurs in a high yield, and a rather obvious formation mechanism can be proposed (Figure 3). In solutions in DCM, the butylene ring of the chloronium cation (CH_2_CH_2_)_2_Cl^+^ breaks, followed by its isomerization to tertiary chlorobutyle^+^, which, by releasing HCl, transitions into the vinyl isobutylene^+^ cation. Subsequently, the two nascent cations, chain chlorobutylene^+^ and isobutylene^+^, interact with a water molecule, thereby generating the final product: protonated ether **I**.

In the IR spectrum of the salt of cation **I,** there is a very intense band of C=C stretch at 1706 cm^−1^, whose frequency exceeds that of the neutral (CH_3_)_2_C=CHOH molecule (Figure 5). Obviously, the positive charge from the OH^+^ group almost does not reach the C=C bond. Furthermore, vibration frequencies of CH_3_ groups (2964 and 2871 cm^−1^) and of C–Cl at 716 cm^−1^ match those of the neutral molecules [17]. Therefore, the positive charge is actually situated on the OH^+^ group with the νOH frequency at 3087 cm^−1^. 

The formation of cation **II** from divinylchloronium cation proceeds in a low yield. Obviously, it is one of a number of emerging products, and it is difficult to propose a sufficiently substantiated mechanism for its formation. We failed to obtain an IR spectrum of cation **II**. It should differ from the spectrum of **I** only in the frequencies of C–Cl and CC bonds of 1,1-dichlorobutylene.

Cations **I** and **II** are not carbocations but protonated esters, which were produced by the interaction of vinyl carbocations with water molecules. In fact, they are the product of the addition of an alcohol molecule to an isobutylene carbocation, as shown in Equation (2) using cation **I** as an example.

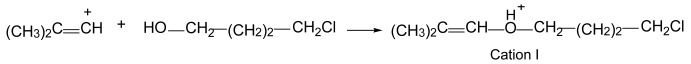
(2)

The formation of carbocations from cations **I** and **II** can be formally represented as the elimination of an H_2_ molecule from the latter, as shown for cation **I** in Figure 4.

Whether such O-containing carbocations exist and whether they can be obtained will be determined by future research.

## 4. Methods and Materials

To obtain the C_4_H_5_Cl_2_^+^{Cl_11_^−^} salt, we used 1,3-dichloro-2-(chloromethyl)-2-methylpropane (98%, Sigma-Aldrich, St. Louis, MO, USA) without further purification, as well as carborane acid H(CHB_11_Cl_11_), which was prepared as described previously [18]. Details of the preparation of the C_4_H_5_Cl_2_^+^{Cl_11_^−^} salt are given in Section 2, as they are necessary to describe the results of the work. In the body of Section 2, details of preparation of protonated ester salts **I** and **II** are also given. The salts of cyclobutylchloronium (CH_2_CH_2_)_2_Cl^+^ and divinylchloronium (CH_2_=CH)_2_Cl^+^ cations used for their preparation were obtained as described previously [12,13]. Solvent C_6_HF_5_ from Sigma-Aldrich was not subjected to additional purification.

All sample handling was carried out in an atmosphere of argon (H_2_O, [O_2_] < 0.5 ppm) in a glove box. ATR IR spectra were recorded on a Shimadzu IRAffinity-1S spectrometer housed inside the glove box in the 4000−400 cm^−1^ frequency range using an ATR accessory with a diamond crystal. The spectra were processed in the GRAMMS/A1 (7.00) software from Thermo Scientific, Waltham, MA, USA.

X-ray diffraction data were collected on a Bruker Kappa Apex II CCD diffractometer using φ,ω-scans of narrow (0.5°) frames with Mo Kα radiation (λ = 0.71073 Å) and a graphite monochromator at temperature 200 K. The structures were solved using direct methods with the help of SHELXT-2014/5 [19] and refined using the full-matrix least-squares method against all F2 in an anisotropic–isotropic (for H atoms) procedure using SHELXL-2018/3 [19]. Absorption corrections were applied using the empirical multiscan method in the SADABS software [20]. Hydrogen atom positions were calculated using the riding model.

The crystallographic data and details of the refinements for all structures are summarized in Appendix A. 

The independent part of the unit cell of the crystal lattice of salts of cations 1,1-dichlorobutylene^+^ and **I** includes two anions and two cations (Appendix A). In both independent cations, the C6, C7, C14, and C15 atoms are disordered over two positions with occupancy ratios of 0.63:0.37 (C6’, C7’) and 0.54:0.46 (C14’, C15’); they are perfectly planar with standard deviations from the mean plane of 0.029 and 0.075 Å, respectively. The geometry of the 1,1-dichlorobutylene^+^ and **I** cations is given in Appendix A, as is the average of their two independent positions. The quite high R-factor for **II** can be explained by the very small size of the analyzed crystal (0.005 × 0.01 × 0.30 mm).

## 5. Conclusions

If uncharged hydrocarbons have substituents such as Cl and Ph, which possess weak electron-withdrawing properties, or, similar to alkylsilylium (–SiR_3_), do not possess them at all, then the binding of these substituents to a positively charged center of the vinyl carbocations causes their transformation into strong electron donors, which quench the positive charge on the C=C bonds, taking it upon themselves. The influx of electron density from substituents to the C=C bond is so considerable that it exceeds that of neutral analogs; the C=C bond’s multiplicity exceeds double and can approach triple. Because this bond involves C atoms with sp^1^ and sp^2^ hybridization, it is expected that some δ^−^ charge appears on it. A detailed study on these features of carbocations using quantum-chemical methods is necessary. 

If an O atom is added to the carbocation, then H^+^ can be transferred to it, with almost the entire charge of the cation concentrated on the =O–H^+^ group. Such an entity becomes a protonated O molecule. On the other hand, it is quite possible that, in the future, it will be possible to obtain vinyl carbocations containing an unprotonated O atom.

Thus, the incorporation of any substituents into vinyl cations turns them into electron donors. If these are X heteroatoms with sufficiently high basicity that allows them to be protonated, then protonated molecules containing the XH^+^ group are generated with a high probability.

## Data Availability

CCDC 2258356, 2258357, and 2258358 contain the supplementary crystallographic data for this paper. These data can be obtained free of charge from The Cambridge Crystallographic Data Center at http://www.ccdc.cam.ac.uk/data_request/cif (accessed on 10 May 2023).

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
