# Peer review of "Substitution of H Atoms in Unsaturated (Vinyl-Type) Carbocations by Cl or O Atoms"

_ijms, 2023, doi:10.3390/ijms241310734_

Round 1

Reviewer 1 Report

Stoyanov and colleagues have presented a study on three structures that are identified as formal addition products of vinyl cations with chloride anions and alcohols. Vinyl cation chemistry has recently garnered significant attention due to its emerging applications in synthetic organic chemistry. The authors have successfully obtained these products through indirect methods, rather than from corresponding vinyl cation salts. The characterization of the structures has been performed through solid state techniques such as IR spectroscopy and single crystal X-ray diffraction.

While the technical execution of the study is commendable, I find it difficult to fully agree with the notion of associating these structures directly with vinyl cations. In my opinion, discussing them solely within the context of vinyl cation chemistry appears to be a slight stretch. Nevertheless, both the salts of dichloroallyl cation and protonated enol ethers are interesting from both structural and reactivity perspectives.

Considering the aforementioned points, I would recommend revising this manuscript more extensively. Shifting the focus of the work away from vinyl cations could enhance its relevance and avoid potential confusion for non-experts in the field. It is worth noting that formally categorizing these molecules as adducts of vinyl cations might be technically correct, but it holds little practical significance in understanding the actual chemistry of vinyl cations.

To illustrate this, we can draw a parallel with the salt of protonated aniline, which could also be viewed as an addition product of NH3 to a phenyl cation. However, this compound is essentially inconsequential in the broader context of phenyl cation chemistry.

Author Response

Response to Reviewer 1.

Stoyanov and colleagues have presented a study on three structures that are identified as formal addition products of vinyl cations with chloride anions and alcohols.

It should be clarify that these are products of the formal combination of vinyl cations with chloride atoms (not anions) and alcohol molecules.

Vinyl cation chemistry has recently garnered significant attention due to its emerging applications in synthetic organic chemistry. The authors have successfully obtained these products through indirect methods, rather than from corresponding vinyl cation salts. (Is it possible to carry out a direct reaction of substitution of the H atom in the vinyl cation for a chlorine atom? E.S.). The characterization of the structures has been performed through solid state techniques such as IR spectroscopy and single crystal X-ray diffraction.

For preparation O-containing cations, cyclobutylchloronium (CH2CH2)2Cl+ and divinyl-chloronium (СН2=СН)2Cl+ salts were used, because they are precursors for obtaining vinyl and acetylene carbocations (there are 5 references to published articles). Therefore, the experimental method for obtaining O-containing cations in this work cannot be indirect. (I cannot understand the meaning of the phrase "indirect method of experimental synthesis").

While the technical execution of the study is commendable, I find it difficult to fully agree with the notion of associating these structures directly with vinyl cations. In my opinion, discussing them solely within the context of vinyl cation chemistry appears to be a slight stretch. Nevertheless, both the salts of dichloroallyl cation and protonated enol ethers are interesting from both structural and reactivity perspectives.

The simplest vinyl cation is C2H3+. According to Wikipedia, all other alkyl-substituted cations with a “+” charge on the C atom at the double bond R2C=C+R are called vinyl, and at the C atom adjacent to the double bond R2C=CR-C+R2  - allyl cations. Such a definition cannot be called strict or generally accepted, especially if the cation contains substituents.

Thus, in the article "The X-ray Structure of a Vinyl Cation" published in Angewandte Chemie, 43, 2004, 1543, the structure of the stabilized vinyl cation R'-C+=C(SiMe2-)2 is discussed in which the nature of the C+=C bond is close to triple. And this cation is called vinyl, although there is practically no “+” charge on the formally double C=C bond. Or in the article "Stabilized vinyl cations" Acc. Chem. Res. 1976, 9, 364 and in many other articles all carbocations containing formally double С=С bonds are called vinyl.

In the carbocations studied by us earlier (all results have been published), it was shown that the nature of the C=C bond can vary from one and half bond status to close to triple, depending on the substituents and the influence of the environment. Therefore, I do not quite understand the wish of the Reviewer to consider the cations discussed in the work not in the context of vinyl cations. Does this only apply to chlorine-containing cations? In the present manuscript we refer to them as chlorobutylene carbocations. (Possibly, the wish of the Reviewer coincides with our opinion on cations I and II, which, as it is written in the manuscript, are formally not vinyl cations, but protonated ethers).

Considering the aforementioned points, I would recommend revising this manuscript more extensively. Shifting the focus of the work away from vinyl cations could enhance its relevance and avoid potential confusion for non-experts in the field. It is worth noting that formally categorizing these molecules as adducts of vinyl cations might be technically correct, but it holds little practical significance in understanding the actual chemistry of vinyl cations.

In all published articles on carbocations containing C=C bonds, such cations are called vinyl cations. This is already established among authors working in this field. If in the manuscript the focus of work is shifted from vinyl carbocations to others, then to which ones? It is impossible to formally classify the discussed cations as adducts of vinyl carbocations, because they are not adducts. We studied adducts of vinyl cations with water and alcohol molecules and published in Molecules 2023, 28, 1146.

 To illustrate this, we can draw a parallel with the salt of protonated aniline, which could also be viewed as an addition product of NH3 to a phenyl cation. However, this compound is essentially inconsequential in the broader context of phenyl cation chemistry.

We propose, as a good compromise, to replace the words “vinyl carbocations” with “vinyl-type carbocations” in the title of the article. Also, wherever possible, the words "chlorinated vinyl carbocations" have been replaced by "chlorinated vinyl-type carbocations". We hope that this is exactly what should satisfy the Reviewer.

Reviewer 2 Report

In my opinion, the paper is an excellent scientific work, carefully it made an analysis of the reaction mechanism from excellent experiment data. Therefore, I suggest that the paper Addition of Cl- and O- atoms to vinyl carbocations  should be published in present form in the International Journal of Molecular sciences.

Author Response

I am happy that Reviewer 2 highly rated the article and has no critical comments

Round 2

Reviewer 1 Report

I have doubts about the author's arguments regarding the connection between these structures and vinyl carbocations. In this work, I see one structure of dichloroallyl cation and two structures of protonated vinyl ethers. Therefore, it would be more appropriate to discuss these structures in relation to allyl cations and protonated vinyl ethers. Unfortunately, I cannot recommend publishing this work in its current state.

Additionally, I would like to provide some general comments. Since there isn't much existing research on these types of compounds, it would be helpful to include detailed experimental procedures for the syntheses, including information on quantities (in moles, milligrams, milliliters, etc.) and yields. Furthermore, the reaction schemes and figures in this work look outdated and of poor quality. The way organic molecules are depicted shows incorrect bond angles.

Author Response

I have doubts about the author's arguments regarding the connection between these structures and vinyl carbocations. In this work, I see one structure of dichloroallyl cation and two structures of protonated vinyl ethers. Therefore, it would be more appropriate to discuss these structures in relation to allyl cations and protonated vinyl ethers. Unfortunately, I cannot recommend publishing this work in its current state.

Reply: This remark surprises me, since it does not correspond to reality. We do not classify the three cations discussed in this work as vinyl cations. The first is related to dichlorobutylene carbocation (page 4), the second and third to protonated ethers (pages 5-7). Let the Referee confirm his statement by citing a sentence from the text where we refer to these cations as vinyl cations!

The purpose of our work is: to study how the isobutylene (vinyl) carbocation changes if the H atoms at different C atoms are replaced in it by 1-2 Cl atoms or by the O atom. Taking into account the previously published results, we come to the conclusion that when replacing one H -atom per Cl-atom, vinyl chlorobutylene cations a-c are formed (Scheme 1 on page 5, these data are published and references are given), allyl c (this work), and intermediate d cation (published). When the O atom is introduced, protonated unsaturated ethers are formed (this work).

This is exactly what the Reviewer wants for the cations studied in this paper. And what does he disagree with?

The meaning of the title of the article proposed by us “Addition of Cl- and O- atoms to vinyl-type carbocations” is the following: how does the vinyl carbocation (butylene) change if Cl or O atoms are introduced into it.

We decided to change the title of the paper to "Substitution of H atoms in unsaturated (vinyl-type) carbocations by Cl or O atoms". Maybe this title reflects the essence of the work more clearly. (Another version of the title was also considered: "Change of vinyl carbocations when their H atoms are replaced by Cl or O atoms").

Additionally, I would like to provide some general comments. Since there isn't much existing research on these types of compounds, it would be helpful to include detailed experimental procedures for the syntheses, including information on quantities (in moles, milligrams, milliliters, etc.) and yields.

Reply: We added to the method of obtaining of a salt of the dichlorobutylene cation: a few drops of a dichloromethane (DCM) solution of CH3С(CH2Cl)3 was introduced into a freshly prepared solution of acid H{Cl11} (3-5 mg) in DCM (~ 1 ml) in such an amount that the molar ratio of CH3С(CH2Cl)3 to H{Cl11} approaches 1.0. Exact amounts of reagents cannot be given as milligram amounts are used. We did not set out to determine the yield of the reaction, since the weight of the resulting crystals cannot be determined (1 mg or less) and the surface of the crystals is contaminated with viscous products of the ongoing side reactions. To develop a method for purifying crystals from oily products, it is necessary to use much larger amounts of carborane superacids, which is unrealistic for the purposes of this work.The described procedure for obtaining crystals is well reproducible, but it is not intended to obtain a significant amount of purified crystals for preparative use.

In the procedure for obtaining crystals of salts of protonated ethers, we added: "saturated solutions of chloronium salts in C6HF5 were used", since their solubility is determined by the content of trace amounts of dissolved water in C6HF5. It is impossible to determine the concentration of the dissolved chloronium salt in these solutions due to the low concentrations of dissolved substances (less than 1 mg in 1 ml) and a small volume of solutions (1-2 ml).

 Furthermore, the reaction schemes and figures in this work look outdated and of poor quality. The way organic molecules are depicted shows incorrect bond angles.

Reply: Figures 2, 3 and 4 show each X-ray structure with valence angles, along with its schematic (topological) image without bond angles (they are shown side by side), but with the designation of the nature of the CC bonds

Therefore, we consider the use of the same schematic representations of cations in schemes 1, 2 and 3 and in the equations on page 7 to be the most illustrative. In the context of this article, this is the best representation of schemas.

Round 3

Reviewer 1 Report

The authors have improved the manuscript. I think it is suitable for publication.